# The Skin Barrier: A System Driven by Phase Separation

**DOI:** 10.3390/cells14181438

**Published:** 2025-09-15

**Authors:** Fengjiao Yu, Lu Leng, Haowen Wang, Mengmeng Du, Liang Wang, Wenhua Xu

**Affiliations:** 1College of Animal Science and Technology, Qingdao Agricultural University, Qingdao 266109, China; 13173319898@163.com (F.Y.); 13255553573@163.com (L.L.); 19560738612@163.com (H.W.); mengmdu@163.com (M.D.); 2Key Laboratory of Pesticide & Chemical Biology of Ministry of Education, Hubei Key Laboratory of Genetic Regulation and Integrative Biology, School of Life Sciences, Central China Normal University, Wuhan 430079, China; 3Laboratory Technology Innovation, Institute of Regenerative Medicine, Qingdao University, Qingdao 266071, China

**Keywords:** liquid–liquid phase separation, skin barrier, filaggrin, atopic dermatitis, RIPK4, corneoptosis, keratohyalin granules, Hippo pathway

## Abstract

The mammalian epidermis forms a critical barrier against environmental insults and water loss. The formation of its outermost layer, the stratum corneum, involves a rapid terminal differentiation process that has traditionally been explained by the “bricks and mortar” model. Recent advances reveal a more dynamic mechanism governed by intracellular liquid–liquid phase separation (LLPS). This review proposes that the lifecycle of the granular layer is orchestrated by LLPS. Evidence is synthesized showing that keratohyalin granules (KGs) are biomolecular condensates formed by the phase separation of the intrinsically disordered protein filaggrin (FLG). The assembly, maturation, and pH-triggered dissolution of these condensates are essential for cytoplasmic remodeling and the programmed flattening of keratinocytes, a process known as corneoptosis. In parallel, an LLPS-based signaling pathway is described in which the kinase RIPK4 forms condensates that activate the Hippo pathway, promoting transcriptional reprogramming and differentiation. Together, these structural and signaling condensates drive skin barrier formation. This review further reinterprets atopic dermatitis, ichthyosis vulgaris, and Bartsocas-Papas syndrome as diseases of aberrant phase behavior, in which pathogenic mutations alter condensate formation or material properties. This integrative framework offers new insight into skin biology and suggests novel opportunities for therapeutic intervention through biophysics-informed biomaterial and regenerative design.

## 1. Introduction

The skin’s primary function as a barrier is executed by the stratum corneum, a resilient layer of flattened, anucleated corneocytes embedded in a lipid-rich matrix [1]. This structure has long been conceptualized by the “bricks and mortar” model, which, while valuable, does not fully capture the dynamic cellular transformations that precede it [2]. The transition from a living granular keratinocyte to a dead corneocyte—a specialized form of programmed cell death now termed corneoptosis [3,4]—is a rapid and highly orchestrated process initiated by a prolonged elevation of intracellular calcium followed by abrupt intracellular acidification [3] and involves massive cytoplasmic reorganization. This transformation is marked by the sudden appearance and disappearance of large, electron-dense structures known as keratohyalin granules (KGs), first described in detail through early electron microscopy studies [5,6]. For decades, the function of KGs and the mechanism of their swift assembly and disassembly remained a puzzle [7].

At the same time, a primary defect in the epithelial barrier has long been implicated in atopic diseases. This link was strikingly confirmed in 2006 by two landmark studies showing that common, loss-of-function variants in the gene encoding Filaggrin (FLG), the major component of KGs, are the direct cause of the common dry skin condition, ichthyosis vulgaris (IV), and represent the most significant known genetic predisposing factor for atopic dermatitis (AD) [8,9]. These null mutations, carried by up to 9% of people of European origin, confer an exceptionally high risk for developing AD, establishing an unequivocal link between an epidermal structural protein and the pathogenesis of complex inflammatory disease [10,11]. This discovery provided strong support for the “outside-in” hypothesis, where a primary barrier impairment allows for enhanced allergen entry, initiating downstream immune dysregulation and the subsequent “atopic march” to asthma and other allergies [9,12,13].

Recent advances in cell biology have revealed that many membraneless organelles, like KGs, are in fact biomolecular condensates formed via liquid–liquid phase separation (LLPS) [14,15,16]. This physical process, driven by multivalent interactions between macromolecules, provides a powerful framework for understanding how cells can rapidly and reversibly compartmentalize their cytoplasm [17,18]. The discovery that germline P-granules in *C. elegans* behave as liquid droplets, localizing via controlled dissolution and condensation, provided one of the foundational examples of this principle in a biological context [14]. This review synthesizes genetic, cell biological, and biophysical evidence to propose a new paradigm: that the formation, function, and pathology of the epidermal barrier are fundamentally driven by the principles of LLPS. It is argued here that KGs are dynamic, liquid-like condensates whose lifecycle orchestrates terminal differentiation [19,20]. Crucially, structural pathway is integrated with the discovery of a parallel, LLPS-dependent signaling axis involving the kinase RIPK4, which co-regulates differentiation [21]. Finally, the genetic basis of AD, IV, and the developmental disorder Bartsocas-Papas syndrome is reinterpreted as diseases of aberrant phase transitions.

## 2. A Primer on Liquid–Liquid Phase Separation in Biology

LLPS is a physical process by which a homogenous solution of macromolecules demixes into two coexisting phases: a dense, polymer-rich phase and a dilute, polymer-poor phase [22]. This phenomenon, long studied by polymer physicists, has gained immense traction in cell biology over the past decade [23]. The realization that the cell interior is not just a well-mixed bag of molecules but a highly organized, spatially compartmentalized environment has led to the identification of numerous membraneless bodies—including nucleoli, Cajal bodies, and stress granules—as biomolecular condensates [16,24,25]. This realization has spurred major efforts to delineate the function of these ‘biomolecular condensates’ and has highlighted the need for rigorous experimental standards to characterize them both in vitro and in cells [26,27].

The primary molecular drivers of biological LLPS are intrinsically disordered proteins (IDPs) and other multivalent macromolecules like RNA [28,29]. Unlike globular proteins, IDPs lack a stable tertiary structure, instead existing as a dynamic ensemble of conformations [30,31]. This conformational heterogeneity allows them to act as flexible polymers capable of forming extensive networks of weak, multivalent interactions, including electrostatic, cation-π, and hydrophobic interactions [32]. From a biophysical standpoint, LLPS occurs when the sum of these weak interactions between macromolecules (chain-chain) becomes more favorable than their interactions with the surrounding water (chain-solvent), causing the system to demix [23,33]. The resulting phase boundary is exquisitely sensitive to protein concentration, temperature, pH, and ionic strength, allowing the cell to tightly control condensate formation and dissolution [34,35].

The interactions that mediate LLPS are multivalent and hierarchical. This has led to the useful scaffold–client framework, where multivalent “scaffolds” drive the phase separation, while “clients”, which may lack this ability on their own, are recruited into the condensate and contribute to its function [17,26]. The specific composition of a condensate is determined by a “molecular grammar” encoded in the amino acid sequences of its components, dictating which molecules are recruited and which are excluded [36,37].Importantly, cells can actively tune phase transitions through multiple regulatory mechanisms, including post-translational modifications (PTMs) like phosphorylation, which can act as a molecular switch to control condensate assembly or disassembly [38], and changes in the ionic environment [39]. The resulting condensates can adopt diverse structures, from simple liquid droplets to complex, multiphasic, or even hollow vesicle-like compartments, highlighting their versatility in organizing cellular biochemistry [24,40].

This mechanism is a conserved strategy for creating high-performance biomaterials. Spider silk, for example, relies on a similar principle. Recombinantly produced spidroins (spider silk proteins) undergo LLPS to form liquid-like coacervate droplets, a process often initiated by dehydration at an air–water interface. This liquid intermediate state is crucial for the subsequent conformational conversion of the protein into highly ordered, β-sheet-rich structures that crystallize to form the final, exceptionally tough silk fiber [41]. Similarly, the protein elastin phase separates to form a dynamic, liquid-like structure that confers elasticity to tissues [42,43]. Thus, in both skin and silk, LLPS serves as a vital intermediate step, concentrating and organizing disordered proteins in preparation for their assembly into a functional, solid-state material.

## 3. The Architects of the Granular Layer and Beyond

### 3.1. The Filaggrin Family: Master Scaffolds for LLPS

The proteins that form the large granules of stratifying epithelia belong to the epidermal differentiation complex (EDC), a gene cluster rich in S100-fused type proteins located on chromosome 1q21 [4,44,45]. Human filaggrin is synthesized as a massive, insoluble polyprotein precursor, profilaggrin (>400 kDa), which is the primary constituent of KGs [7,46]. The FLG gene has a unique “fused-gene” structure, with a large central exon encoding 10–12 tandem, near-identical filaggrin repeats flanked by N- and C-terminal domains [12,47]. This genetic architecture is complex, featuring intragenic copy number variation (CNV) that results in individuals having between 10 and 12 filaggrin repeats per allele. This CNV itself contributes to eczema risk in a dose-dependent manner, independent of null mutations [12,48]. Bioinformatic analysis reveals that FLG and its paralogs are quintessential IDPs, enriched in disorder-promoting residues like glycine and serine and sharing compositional biases that favor LLPS [5,20,49]. This disordered nature is critical for preventing the formation of rigid, amyloid-like structures, ensuring the granules remain in a dynamic, reversible state [42,50].

This architectural theme is conserved in other EDC proteins. FLG paralogs like Repetin (RPTN) and Filaggrin-2 (FLG2) also form granules in the epidermis [51,52,53]. RPTN can form distinct condensates that are immiscible with KGs, suggesting they have different material properties and functions [20]. FLG2, whose truncating mutations are linked to Peeling Skin Syndrome and AD in African Americans [54,55], appears to colocalize with FLG but may have non-overlapping roles [4]. Similarly, Hornerin is another S100-fused type protein whose deficiency in mouse models, when combined with FLG deficiency, leads to a more severe barrier defect and a lowered allergic threshold, suggesting partially redundant functions [56]. In the inner root sheath of the hair follicle, Trichohyalin (TCHH) is the major component of Trichohyalin Granules (TGs) [57,58]. Like FLG, TCHH is a large, repetitive, S100-fused IDP, but it has a distinct composition rich in arginine and glutamine [4,59]. TCHH functions as a cross-linking protein, mechanically strengthening the hair follicle [60]. Truncating mutations in TCHH disrupt TG formation and cause Uncombable Hair Syndrome, demonstrating that LLPS-driven granule formation is a conserved mechanism across different epithelial appendages [61].

### 3.2. The Keratin Cytoskeleton: A Dynamic Cage

As keratinocytes enter the spinous and granular layers, they assemble a dense network of keratin intermediate filaments (K1/K10) [62]. These filaments are not merely a static scaffold but are highly dynamic structures essential for the mechanical integrity of the epidermis [63]. They possess large, low-complexity domains that mediate a crucial interaction with KGs [64]. Rather than being inert bystanders, the keratin filaments actively organize the cytoplasm by “caging” the liquid-like KGs, restricting their fusion and controlling their size [20,65]. This interaction appears to be mediated by the low-complexity domains of K1/K10 binding to the surface of FLG condensates [20]. This caging mechanism effectively creates an elastic network that imposes physical constraints on the phase-separating FLG, a phenomenon that has been shown in physical models to modulate the thermodynamics of LLPS itself [66]. This interplay ensures that as KGs grow, they do not coalesce into a single, massive droplet but instead form a distributed network of condensates that fills the cytoplasm.

## 4. Keratohyalin Granules: An LLPS-Driven Lifecycle

The work of Quiroz et al. (2020) provided definitive evidence that KGs are not inert aggregates but are dynamic biomolecular condensates whose entire lifecycle—assembly, maturation, and dissolution—is governed by LLPS [20] (Figure 1). This perspective builds upon decades of morphological studies that hinted at a highly dynamic process but lacked a unifying physical mechanism [67,68].

### 4.1. Assembly: A Concentration- and Valency-Dependent Phase Transition

The formation of KGs is a classic example of a concentration-dependent phase transition. FLG must reach a critical concentration to trigger LLPS. This process is exquisitely sensitive to valency—the number of repeating domains. Quantitative experiments in cells, using fluorescently tagged FLG constructs and precisely controlling their expression levels, showed that wild-type FLG (12 repeats) forms droplets at a critical concentration of just ~2 µM [20,69]. In stark contrast, disease-associated variants with four or fewer repeats fail to phase separate under physiological conditions, with critical concentrations skyrocketing to ~130–1500 µM [20]. This provides a direct biophysical explanation for the loss of KGs in patients with severe FLG mutations [70]. The initial step is also thought to be regulated by the N-terminal S100 calcium-binding domain of profilaggrin, which may facilitate the initial self-association in a calcium-dependent manner, linking the process to the known calcium gradient in the epidermis [71,72].

### 4.2. A Crowded, Structured Liquid: Maturation and Cytoplasmic Organization

Once formed, KGs are highly dynamic, liquid-like droplets, as confirmed by fluorescence recovery after photobleaching (FRAP) and atomic force microscopy (AFM) [20]. While these observations strongly support a liquid-like state, the field emphasizes that properties such as sphericity and rapid FRAP recovery should be interpreted cautiously in the complex cellular environment and combined with evidence of concentration-dependent assembly to rigorously define a structure as a product of LLPS [26]. Live imaging in mouse skin revealed that as KGs mature, they become progressively more viscous and gel-like, a process known as aging or hardening that is common in biological condensates [20,73]. This maturation is accompanied by their “caging” within the K1/K10 keratin network, which prevents runaway fusion and allows the cytoplasm to become progressively crowded [20,65]. This increase in macromolecular crowding has profound effects on cellular biochemistry, altering reaction rates and potentially triggering phase separation of other components [74,75]. This crowding exerts significant physical force, deforming the nucleus, a process hypothesized to contribute to its eventual degradation during corneoptosis [19,20]. The mechanical stress from growing KGs may also trigger mechanosensitive pathways that further promote differentiation [76]. This dynamic, hydrated, and disordered but cohesive state is analogous to the ‘liquid structure’ described for elastin, where phase separation also produces a functional biomaterial without forming a static, water-excluding core [42,43].

### 4.3. Dissolution: A Multimodally Regulated Switch for Corneoptosis

The final step is the abrupt dissolution of KGs as keratinocytes transition into the stratum corneum. This disassembly releases a cascade of functional molecules. The profilaggrin polyprotein is dephosphorylated and rapidly cleaved by a series of proteases into multiple monomeric filaggrin units [7,77,78]. Key proteases in this cascade include matriptase, prostasin, caspase-14, and SASPase, which act sequentially to process the large precursor [79,80,81,82,83]. These monomers bind to and aggregate the keratin cytoskeleton, causing the cell to collapse and flatten into a squame [84,85]. Subsequently, filaggrin itself is degraded into a collection of small, hygroscopic amino acids and their derivatives, which constitute the skin’s Natural Moisturizing Factor (NMF), critical for stratum corneum hydration [12,86]. These breakdown products also include acidic molecules like urocanic acid and pyrrolidone carboxylic acid, which help maintain the low pH of the stratum corneum (the “acid mantle”), providing antimicrobial defense and regulating enzymatic activity [12,87].

This entire process, a key step in corneoptosis, is initiated by a rapid drop in intracellular pH that follows a sustained elevation of intracellular calcium lasting approximately 60 min [3]. The work by Matsui et al. (2021) demonstrated that this intracellular acidification is essential for the subsequent degradation of both organelles and nuclear DNA [3]. The degradation of nuclear DNA itself is carried out by specific endonucleases like DNase1L2 and DNase2, which are activated during cornification [88,89]. Providing a direct biophysical mechanism for this observation, the high histidine content of FLG acts as a pH sensor; acidification protonates these residues, increasing net positive charge and electrostatic repulsion, which disrupts the condensate in vitro and likely making the profilaggrin accessible to processing enzymes in vivo [20,90]. In addition to pH, other PTMs like deimination—the conversion of arginine to citrulline by peptidylarginine deiminases (PADs)—play a critical role. Deimination of FLG reduces its positive charge, which is thought to facilitate its release from the keratin network and prepare it for final degradation [91,92,93]. Environmental factors, such as low humidity, can even increase PAD activity, accelerating FLG breakdown and potentially compromising barrier function under dry conditions [94].

Based on these findings, a multi-step model for KG dissolution is emerging, though several aspects remain speculative and require further investigation [4,19]. First, it has been proposed that the S100 domain, which promotes the initial LLPS, is cleaved early after KG formation to maintain the granule’s liquidity, a model supported by in vitro and indirect in vivo evidence [20,95]. Later, at the granular-to-corneum transition, the pH drop acts as the primary trigger. Subsequently, it is hypothesized that this trigger is followed by PTMs, such as phosphorylation, which may further drive disassembly and prevent premature re-aggregation [4,96]. Finally, this model posits that pH-activated proteases like SASPase can begin processing the now-accessible FLG scaffold [3,79].

## 5. The RIPK4-Hippo Axis: A Parallel LLPS-Based Signaling Hub

While the essential role of Receptor-interacting serine/threonine kinase 4 (RIPK4) in keratinocyte differentiation is well-established [97,98], the precise signaling mechanism has only recently been uncovered. RIPK4 belongs to the RIP kinase family, which are key regulators of cellular stress, inflammation, and cell death [99,100]. Intriguingly, long before its connection to the Hippo pathway was known, RIPK4 was independently identified as a direct interacting partner of Protein Kinase C (PKC) isoforms (specifically PKCδ and PKCβ) in keratinocyte models [101,102]. This initial discovery is highly significant, as the PKC family is a central mediator of the calcium signaling that drives epidermal differentiation, providing a direct link between RIPK4 and the primary intracellular cues for this process. Building on this foundation, recent work has elucidated that RIPK4 functions as a LLPS-driven, non-canonical upstream kinase for the Hippo pathway, which is required for proper epidermal development [21]. The Hippo pathway is a highly conserved signaling cascade that plays a central role in controlling organ size by regulating cell proliferation and apoptosis [103,104,105].

Through its ankyrin repeat domains—protein–protein interaction motifs known for their structural versatility [106]—RIPK4 undergoes LLPS to form signaling condensates in the cytoplasm of differentiating keratinocytes [21]. These condensates act as hubs to recruit and concentrate the Hippo pathway kinases LATS1/2, leading to their direct phosphorylation and activation by RIPK4. This represents a non-canonical, differentiation-specific activation of the Hippo pathway, bypassing the canonical upstream kinases MST1/2 [21,107]. Activated LATS1/2 then inhibit the transcriptional co-activators YAP/TAZ specifically in the granular layer [108,109,110]. While YAP/TAZ are known to promote proliferation in the basal layer [111], their RIPK4-mediated inhibition in the granular layer is necessary to suppress a proliferative program and to upregulate genes involved in cholesterol biosynthesis, a key lipid component of the skin barrier [21,112]. This discovery reveals that the skin barrier is regulated by the interplay of at least two distinct LLPS systems: a structural system (FLG/KGs) and a signaling system (RIPK4/LATS) [21] (Figure 2). This parallels findings in other signaling pathways, such as T-cell receptor activation, where phase separation of key components is essential for signal amplification and transduction [113]. The ability of RIPK4 condensates to concentrate kinases and substrates is a classic example of how LLPS can enhance biochemical reactions and create specific signaling outcomes [114].

## 6. Pathophysiology: When Phase Separation Fails

The LLPS paradigm provides a powerful biophysical explanation for the molecular genetics of common and rare skin barrier diseases.

### 6.1. Atopic Dermatitis and Ichthyosis Vulgaris: Diseases of Altered Critical Concentration

The discovery that common loss-of-function mutations in FLG are the primary cause of IV and the strongest known genetic risk factor for AD was a watershed moment [8,9]. These mutations are prevalent worldwide, though the specific alleles vary between populations, with distinct mutations identified in European, Japanese, and Singaporean Chinese cohorts [115,116,117]. From an LLPS perspective, these genetic findings have clear biophysical consequences. Severely truncating mutations dramatically reduce FLG’s multivalency, elevating the critical concentration for phase separation to a point where KGs fail to form [20,70]. However, the story is more nuanced. Milder mutations, such as those deleting the C-terminal tail, do not abolish KG formation but significantly decrease the viscosity of the resulting condensates. These less-viscous KGs “wet” the nuclear surface rather than deforming it, suggesting that the specific material properties of KGs, not just their presence, are critical for proper function [20]. Furthermore, the dose-dependent effect of FLG CNV on AD risk highlights the quantitative nature of this phase transition; a lower copy number translates to a lower overall protein concentration, moving the cell closer to the phase boundary and making it more susceptible to barrier failure [48]. Therefore, AD and IV can be re-conceptualized not just as diseases of FLG absence, but as disorders of altered phase boundaries and aberrant material properties (Figure 3).

### 6.2. Bartsocas-Papas Syndrome: A Disease of Defective Signaling Condensates

The principles of LLPS-driven barrier formation find a compelling parallel in other genetic syndromes. Bartsocas-Papas syndrome, a severe developmental disorder, is caused by inactivating mutations in RIPK4 [118]. Similar mutations in RIPK4 are also responsible for the autosomal-recessive form of popliteal pterygium syndrome, underscoring its critical role in epithelial development [119]. Recent work has shown that disease-derived RIPK4 mutants are defective in their ability to activate LATS1/2 because they either lack kinase activity or are unable to undergo LLPS due to truncation of the ankyrin repeat domain [21]. This positions Bartsocas-Papas syndrome as a disease caused by the failure of a signaling condensate, highlighting the importance of LLPS in both structural and regulatory aspects of skin development. This failure prevents the proper suppression of the proliferative YAP/TAZ program in the upper epidermal layers, leading to the severe developmental defects characteristic of the syndrome [21,111].

## 7. Conclusions

The conceptual leap provided by LLPS has resolved the long-standing enigma of keratohyalin granules. Skin barrier formation is not merely a sequence of gene expression events, but a process fundamentally governed by the physical chemistry of phase separation. This process is at least twofold: structural LLPS driven by FLG organizes the cytoplasm, while signaling LLPS driven by RIPK4 co-regulates the differentiation program.

This new paradigm opens exciting avenues for future research. Four major directions stand out: First, understanding the environmental resilience of the barrier. How do external factors like temperature and humidity, which are known to exacerbate AD, directly impact the LLPS dynamics of KGs [120,121]? Given that FLG exhibits temperature-sensitive phase behavior in vitro, its condensates in the outer epidermis may be exquisitely tuned to respond to environmental shifts [4,34]. This concept of environmental sensing is supported by recent findings in other keratinocytes derived from the oral mucosa. A study demonstrated that the disassembly and reassembly of cytoplasmic condensates in response to hypotonic and thermal stress are temperature-dependent and regulated by the WNK-SPAK/OSR1 kinase pathway. This provides a specific molecular link between environmental stimuli and condensate dynamics, highlighting that additional signaling cascades are likely involved in tuning phase behavior to maintain epithelial homeostasis [122]. Gene–environment interactions are critical, as studies have shown that factors like early-life cat exposure can enhance the eczema risk conferred by FLG mutations [12]. Second, dissecting the role of intracellular crowding and organelle interactions. The interplay between growing KGs, the keratin network, and membrane-bound organelles is a key area. During cornification, organelles must be cleared in a coordinated fashion. Mitochondrial fragmentation, driven by proteins like NIX and DRP1, is an early step in this process [123]. Concurrently, autophagy is induced to degrade cellular contents, a process essential for terminal differentiation [124]. The link between KG dissolution and organelle degradation during corneoptosis is becoming clearer, with intracellular acidification acting as a common trigger [3]. How KG-induced crowding primes these organelles for degradation remains an open question. Third, identifying the full suite of molecular modulators. The roles of PTMs, FLG paralogs (RPTN, FLG2), and other IDPs like loricrin in tuning KG phase behavior remain largely unexplored. Loricrin, for instance, is an IDP that colocalizes with KGs in human skin and could be a key client or modulator [4,125]. Furthermore, filaggrin deficiency can be acquired, as inflammatory cytokines such as IL-4 and IL-13 can down-regulate FLG expression, creating a vicious cycle where inflammation further compromises the barrier [12,126]. Fourth, exploring LLPS beyond granules. The principle of phase separation may extend to other epidermal structures, such as the assembly of tight junctions [4,127].

Ultimately, understanding the precise biophysical tuning of these LLPS processes offers the potential for novel therapeutic strategies. This bio-inspired approach is already being explored in materials science for applications in drug delivery [128], wound healing [129], and skin electronics [130]. Similarly, LLPS-based biomaterials have been shown to accelerate skin repair [131]. These studies provide a powerful proof-of-concept for designing materials that can create pro-regenerative microenvironments, moving beyond simply suppressing inflammation to actively repairing the barrier at a fundamental, biophysical level. Furthermore, the relevance of LLPS in skin biology may extend beyond barrier function, with emerging evidence linking aberrant phase separation to the progression of skin cutaneous melanoma, suggesting new therapeutic targets for skin cancers [132].

## Figures and Tables

**Figure 1 cells-14-01438-f001:**
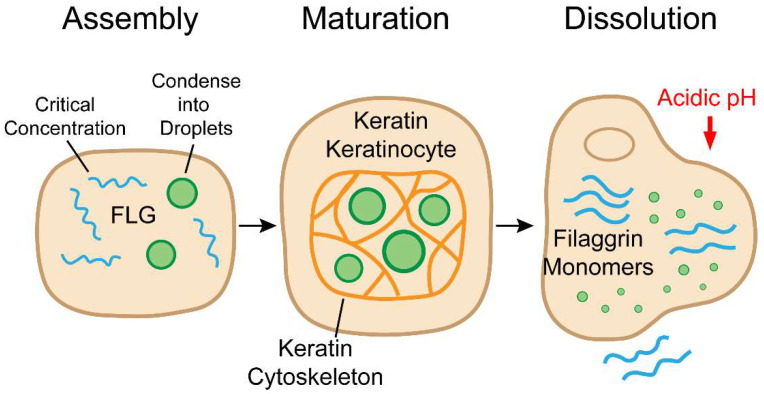
The Lifecycle of Keratohyalin Granules (KGs) Driven by LLPS. Keratohyalin granules (KGs) in granular keratinocytes undergo a dynamic, three-stage lifecycle orchestrated by liquid–liquid phase separation (LLPS). Left, upon reaching a critical concentration, intrinsically disordered filaggrin (FLG) proteins undergo LLPS to form liquid droplets. Middle, these droplets mature into more viscous condensates constrained by the keratin cytoskeleton (K1/K10), organizing the cytoplasm and priming the nucleus for destruction. Right, a drop in intracellular pH triggers dissolution of the granules, releasing FLG monomers that aggregate keratin filaments, drive cell flattening (corneoptosis), and ultimately generate natural moisturizing factor (NMF) components.

**Figure 2 cells-14-01438-f002:**
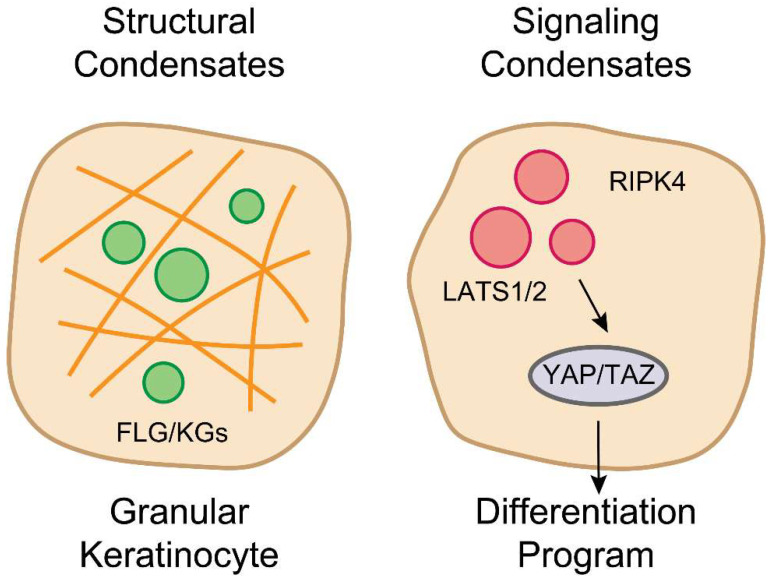
Two LLPS Systems Coordinate Epidermal Differentiation: Structural vs. Signaling Condensates. Epidermal differentiation is regulated by two distinct LLPS-driven systems within granular keratinocytes. Left, structural Condensates: FLG forms keratohyalin granules (KGs), which are stabilized by keratin filaments and remodel the cytoplasm. Right, signaling Condensates: The kinase RIPK4 forms red signaling droplets that recruit and activate LATS1/2 kinases, which in turn inhibit YAP/TAZ transcriptional co-activators. This repression initiates the differentiation program, illustrating a parallel LLPS-dependent signaling axis. Together, these condensates orchestrate the structural and transcriptional events necessary for skin barrier formation.

**Figure 3 cells-14-01438-f003:**
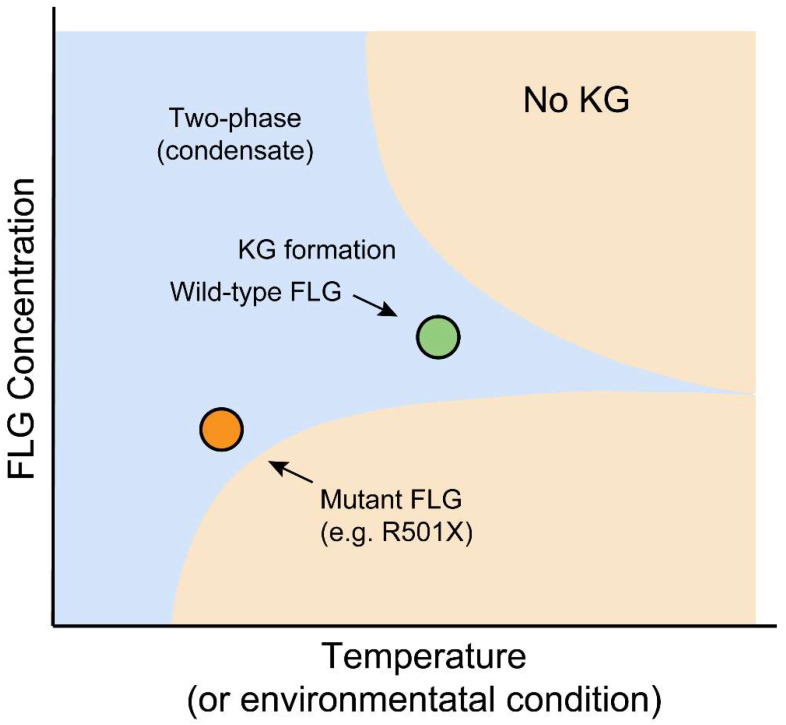
Pathogenic Shift in Critical Concentration for FLG Phase Separation. A phase diagram illustrating the effect of FLG truncation mutations on LLPS behavior. The binodal curve separates the two-phase (condensate-forming) and one-phase (soluble) regions. Wild-type FLG, due to its high multivalency, undergoes LLPS at physiological concentrations, driving KG formation. Mutant FLG variants (e.g., R501X), with reduced repeat numbers, exhibit a rightward shift in the saturation concentration (Csat), placing physiological levels within the one-phase region and abolishing condensate formation. This biophysical failure underlies skin barrier dysfunction in ichthyosis vulgaris and atopic dermatitis.

## Data Availability

No new data were created or analyzed in this study. Data sharing is not applicable to this article.

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
