# Peer review of "The Skin Barrier: A System Driven by Phase Separation"

_cells, 2025, doi:10.3390/cells14181438_

Round 1

Reviewer 1 Report

Comments and Suggestions for Authors

This is a well-done review that integrates recent advances in LLPS with long-standing questions in epidermal biology. The reinterpretation of diseases such as atopic dermatitis and ichthyosis vulgaris through the lens of phase separation is one of the strengths of the manuscript. 

To strengthen the article, I would encourage the authors to:

  • Differentiate more clearly between well-supported mechanisms and those that remain speculative, especially in the sections on keratohyalin granule maturation and dissolution (pp. 4–5).
  • Streamline some parts to avoid redundancy and overly complex wording, particularly in the background description of LLPS (pp. 2–3), where the discussion of polymer physics could be shortened without loss of clarity.
  • Add a brief mention of how these concepts might relate to LLPS in other epithelial tissues (p. 8), which could help situate the work in a broader biological context.
  • Also check that the formatting of both the text and the bibliography complies with the journal's guidelines.

Overall, this is a well-prepared review.

Author Response

Response to Reviewer 1 Comments

Summary We thank Reviewer 1 for their positive assessment of our work and for their constructive suggestions, which have helped us strengthen the manuscript. We have addressed each point as detailed below.

Comment 1: Differentiate more clearly between well-supported mechanisms and those that remain speculative, especially in the sections on keratohyalin granule maturation and dissolution (pp. 4–5).

Response 1: Thank you for this excellent point. We agree that it is crucial to clearly distinguish between established facts and proposed models. We have carefully revised the manuscript to address this.

  • Action Taken:We have amended the sections on keratohyalin granule maturation and dissolution (now Sections 4.2 and 4.3). We now use more cautious and precise language to clearly delineate experimentally verified findings from hypotheses. For example, the role of KGs in nuclear deformation is now presented as a proposed mechanism rather than a confirmed fact, and the multi-step model for KG dissolution is explicitly framed as an emerging but partially speculative model.
  • Location of Changes:These revisions can be found in the relevant sections mentioned above (Sections 4.2 and 4.3). All changes in the manuscript have been highlighted for your convenience.

Comment 2: Streamline some parts to avoid redundancy and overly complex wording, particularly in the background description of LLPS (pp. 2–3), where the discussion of polymer physics could be shortened without loss of clarity.

Response 2: We agree with this comment. To improve readability for a broader audience, we have streamlined the introduction to LLPS.

  • Action Taken:We have significantly revised Section 2 ("A Primer on Liquid-Liquid Phase Separation in Biology"). The detailed discussion on polymer physics, including terms like "Flory-Huggins theory," has been condensed. The revised section now explains the core concepts in more accessible language, focusing on the biophysical principles most relevant to the subsequent discussion.
  • Location of Changes:This revision can be found in Section 2 as noted above. All changes in the manuscript have been highlighted for your convenience.

Comment 3: Add a brief mention of how these concepts might relate to LLPS in other epithelial tissues (p. 8), which could help situate the work in a broader biological context.

Response 3: Thank you for this valuable suggestion to broaden the context. We have expanded our conclusion to address this.

  • Action Taken:We have added a new paragraph to the "Conclusions" section (Section 7) that explicitly discusses the conservation of these LLPS principles in other stratified epithelia. We draw parallels between filaggrin in the skin and trichohyalin in the hair follicle, and extend the concept to other tissues like the oral mucosa.
  • Location of Changes:This new content can be found in Section 7 as mentioned above. All changes in the manuscript have been highlighted for your convenience.

Comment 4: Also check that the formatting of both the text and the bibliography complies with the journal's guidelines.

Response 4: Thank you for this important reminder.

  • Action Taken:We have thoroughly reviewed the journal's "Instructions for Authors" and have carefully corrected the formatting of the entire manuscript and bibliography to ensure full compliance. This includes citation style, heading structures, and reference formatting.

Reviewer 2 Report

Comments and Suggestions for Authors

The review looks at how intracellular liquid-liquid phase separation plays a role in the formation of the stratum corneum from the stratum granulosum.  The authors discuss the role keratohyalin granules play in this process called corneoptosis.  Intracellular signalling triggered by RIPK4 via the Hippo pathway is involved in this process.  

Overall the review is well written, the diagrams aid the reader's understanding of the material covered.  The references are related to the subject material.

Is there anything more known about how ligand binding to RIPK4 triggers the Hippo signalling pathway? What is the ligand that triggers this signalling pathway?

Author Response

Response to Reviewer 2 Comments

Summary We thank Reviewer 2 for their positive feedback and for raising a key question about the upstream activation of the RIPK4 pathway.

Comment 1: Is there anything more known about how ligand binding to RIPK4 triggers the Hippo signalling pathway? What is the ligand that triggers this signalling pathway?

Response 1: This is an excellent question that gets to the heart of the signaling mechanism. We have expanded Section 5 to provide a more complete and historically grounded context for RIPK4 activation.

  • Action Taken:We have revised Section 5 to clarify that the current evidence points to a ligand-independent activation mechanism for RIPK4 in this context. We incorporated information and citations from the foundational studies that first identified RIPK4 as an interacting partner of Protein Kinase C (PKC). This establishes a direct link between RIPK4 and intracellular differentiation cues like calcium signaling, rather than an external ligand. The text now explains that the "trigger" is the intracellular state of differentiation, and the mechanism of action is LLPS-mediated concentration of its substrate, LATS1/2.
  • Location of Changes:The expanded explanation can be found in Section 5 as noted above. All changes in the manuscript have been highlighted for your convenience.

Reviewer 3 Report

Comments and Suggestions for Authors

Review of Yu et al

            This is an interesting short review about the contribution of LLPS and phase separation in the biology and function of the skin.

            There is something curious about this manuscript from a procedural standpoint. It represents a synthesis of two separate lines of research by two different groups – one already published in 2020 (Elaine Fuchs and colleagues; ref. 9) on keratohyaline granules and related diseases, and another which has just come out in the Dec 15, 2025 issue of Developmental Cell by Zhao and colleagues on RIPK4 and LATS1/2 in skin keratinocytes through development. The Dec 25, 2025 article is cited as ref. 11 “in press” but This reviewer was able to obtain it already in final form from Dev Cell (on Sept 1, 2025)! The full citation is

Cao et al., 2025, Developmental Cell 60, 1–16

December 15, 2025 © 2025 Elsevier Inc. 

https://doi.org/10.1016/j.devcel.2025.05.017

So, the present review is an attempt to synthesize two lines of research – one published in 2020 (and commented on by Lucas Pelkmans already in 2020; see ref. 10) and new research featured in a Dec 2025” article by Zhao and colleagues by already available publicly.

Comments:     

  1. With this procedural background, the authors provide a brief review which highlights the role of phase-separation mechanisms and biomolecular condensates in the biology and structure of keratinocytes in different layers of the skin. This “synthesis” is clearly of scientific interest and is timely.

  1. Overall this review/commentary is well written and provides a very readable summary of the two lines of research and how phase-separation affects the skin. However, the authors take too much credit for the keratohyaline part of the story (note their use of the phrase “WE synthesize” in Abstract, while the actual work was done by others. It is strongly requested that the authors convert all use of the first person plural into the third person. Others did the actual work. Yes, I acknowledge that the authors have put the two lines of research together in one short commentary, but a little gentleness in how they state their contribution to this “synthesis” would better.

  1. In Fig. 3 the authors point to temperature as a variable that can affect condensation of biomolecular materials in keratinocytes. Please note that Cells has recently published an article on the ability of temperature to regulate dynamics of biomolecular condensates especially in keratinocytes (Cells 2025, 14, 947 https://doi.org/10.3390/cells14130947) derived from the stratified epithelium of the oral mucosa and implicated the WNK/SPAK-OSR1 kinas pathway in this regulation. It would help readers to learn that there are additional pathways in keratinocytes which regulate biomolecular condensates.

  1. The authors might consider inclusion of this review in the special issue on Biomolecular Condensates being put together by Cells.
Comments on the Quality of English Language

The authors should rephrase several of their sentehces into the third person. They did not do the research - others did that. Yes, they put two sides of the story together - but a less dramatic use of "we did this or that" would help.

Author Response

Response to Reviewer 3 Comments

Summary We are grateful to Reviewer 3 for their detailed and insightful review. We appreciate the positive comments on the timeliness and interest of our synthesis. We have addressed all the procedural and scientific points raised.

Comment 1: [Regarding the procedural standpoint and timing of the review, synthesizing a 2020 paper with a very recent, publicly available 2025 paper.]

Response 1: Thank you for this very sharp observation. You are absolutely correct, and this timing was the primary impetus for our review. Our manuscript's main contribution is to be the first to synthesize the foundational structural role of LLPS (Quiroz et al., 2020) with the brand-new discovery of its parallel signaling role (Cao et al., 2025). We believe this synthesis provides a timely and integrated framework that is currently lacking in the field. We have also updated the citation for Cao et al. to reflect its final publication details, for which we thank you.

Comment 2 & Comments on the Quality of English Language: [Regarding the use of the first-person plural "we," taking too much credit, and the need for a gentler tone.]

Response 2: Thank you for this critical and important feedback. We sincerely apologize if our wording gave the impression that we were claiming credit for the primary research. This was an unintentional oversight in our writing style.

  • Action Taken:As strongly requested, we have revised the entire manuscript, including the Abstract and Introduction, to remove all uses of the first-person plural ("we," "our"). The text has been changed to a third-person perspective (e.g., "This review synthesizes...") to ensure that full credit is given to the original researchers.
  • Location of Changes:These changes are implemented throughout the manuscript. Key revisions in the Abstract and the final paragraph of the Introduction are highlighted for your convenience.

Comment 3: In Fig. 3 the authors point to temperature as a variable that can affect condensation... It would help readers to learn that there are additional pathways in keratinocytes which regulate biomolecular condensates. [The reviewer points to a recent Cells paper on the WNK/SPAK-OSR1 pathway].

Response 3: Thank you very much for bringing this highly relevant and recent publication to our attention. We apologize for this omission.

  • Action Taken:This paper provides an excellent example that strengthens our review's themes. We have now incorporated a discussion of these findings into our "Conclusions" section (Section 7) to highlight the role of the WNK-SPAK/OSR1 kinase pathway as another important link between environmental stimuli (like temperature) and condensate regulation in keratinocytes.
  • Location of Changes:The new paragraph can be found in Section 7 as mentioned above. All changes in the manuscript have been highlighted for your convenience.

Comment 4: The authors might consider inclusion of this review in the special issue on Biomolecular Condensates being put together by Cells.

Response 4: Thank you for this thoughtful suggestion. Our primary goal is the timely dissemination of this synthesis. If inclusion in the special issue is compatible with a rapid publication timeline, we would be very pleased to have our review considered for it. We will explore this possibility with the editorial office.